# Effects of Myofascial Release Techniques on Joint Range of Motion of Athletes: A Systematic Review and Meta-Analysis of Randomized Controlled Trials

**DOI:** 10.3390/sports12050132

**Published:** 2024-05-14

**Authors:** Bogdan Alexandru Antohe, Osama Alshana, Hüseyin Şahin Uysal, Marinela Rață, George Sebastian Iacob, Elena Adelina Panaet

**Affiliations:** 1Departament of Physical Therapy and Ocupational Therapy, “Vasile Alecsandri” University of Bacău, 600011 Bacău, Romania; antohe.bogdan@ub.ro (B.A.A.); panaet.adelina@ub.ro (E.A.P.); 2Department of Physiotherapy, University College of Applied Sciences, Gaza Strip P6160675, Palestine; osamashana20@gmail.com; 3Faculty of Sport Sciences, Burdur Mehmet Akif Ersoy University, 15030 Burdur, Turkey; 4Department of Physical Education and Sports Science, Faculty of Physical Education and Sport, “Alexandru Ioan Cuza” University of Iași, 700506 Iași, Romania; georgesebastianiacob@gmail.com

**Keywords:** myofascial release therapy, joint range of motion, athletes, athletic performance, sports

## Abstract

Although myofascial release techniques (MRTs) are commonly used to improve athletes’ range of motion (ROM), the effectiveness of MRTs may vary depending on the specific method performed. This systematic review and meta-analysis aimed to evaluate the effects of MRTs on the ROM performance of athletes. (2) Methods: The electronic databases of Cochrane Library, PubMed, Scopus, and Web of Science were searched to identify relevant articles published up to June 2023. This study utilized the PRISMA guidelines, and four databases were searched. The methodological quality of the studies was assessed using the PEDro scale, and the certainty of evidence was reported using the GRADE scale. The overall effect size was calculated using the robust variance estimator, and subgroup analyses were conducted using the Hotelling Zhang test. (3) Ten studies met the inclusion criteria. The overall effect size results indicated that the myofascial release intervention had a moderate effect on ROM performance in athletes when compared to the active or passive control groups. (4) Conclusions: Alternative MRTs, such as myofascial trigger point therapy, can further improve the ROM performance of athletes. Gender, duration of intervention, and joint type may have a moderating effect on the effectiveness of MRTs.

## 1. Introduction

Joint range of motion (ROM) refers to the capability of a joint to go through its complete spectrum of movements. An optimal ROM is essential for maintaining athletic performance and preventing injury. Therefore, practitioners, coaches, and physiotherapists aim to increase or maintain the ROM of athletes [1]. ROM can impact various fitness components that require strength, power, sprint speed, and vertical jump [2,3]. In addition, the ROM level may also be necessary for clinical evaluation of the musculoskeletal system, maintenance of athletes’ performance, and injury prevention evaluations [4,5]. When musculoskeletal ultrasonography and ROM evaluation are used together, they can be a more effective and reproducible diagnostic tool [4]. Previous studies have reported that an optimal ROM is necessary to maximize the effectiveness of resistance training [6,7]. For example, ten weeks of resistance training with a full ROM increased maximal strength performance compared to resistance training with a partial ROM in male athletes by 16–27% [6]. Similarly, researchers observed a large effect size (ES) for increased ROM in explosive tasks when evaluating the impact of ROM on post-activation performance improvement [7]. Furthermore, a previous study showed that ROM can increase stride length in sprinters [8]. The study found a high correlation between lower limb ROM and start sprint performance [8]. In another study, it was suggested that achieving a full ROM can improve the maximal sprint speed of team athletes [9]. Empirical evidence demonstrated a strong correlation between lower limb ROM and vertical jump performance among elite handball players [10].

An optimal ROM can enhance the performance of athletes. However, the ROM of an athlete may be restricted by sport-specific activity (e.g., gymnasts have a better ROM than powerlifters) [11], biological factors (gender, hormones, age) [12], anthropometric characteristics (weight, height, shape of the articular surfaces [13], or physical activity level. Therefore, various techniques are used to improve the ROM of the athlete. Myofascial release is a technique that can be used to improve the ROM by releasing tension in the fascia [14]. Myofascial release can help prevent pain and discomfort caused by tight muscles [15], and it can also improve circulation [14].

Foam rolling, stretching, and proprioceptive neuromuscular facilitation (PNF) are some techniques used for myofascial release. The studies that have evaluated the effects of myofascial release techniques on ROM have reported conflicting results over the past few years. For example, one study pointed out that using a foam roller on the lower limbs for more than 90 s had a small effect on ROM. However, when the foam roller was used for less than 90 s, there was no significant effect on lower limb ROM performance [16]. A systematic review and meta-analysis evaluated the effects of stretching, foam rolling, and combined therapy techniques (i.e., foam rolling and stretching) on ROM performance. The findings indicated that combined therapy had a small positive effect size on improving ROM compared to the control group [16]. However, there was no significant effect of the combined therapy compared to the stretching and single foam roller groups [17]. Also, a recent systematic review and meta-analysis found that vibrating foam rolling was more effective than standard foam rolling in improving the ROM of the participants [18]. Researchers have argued that stretching enhances ROM performance depending on the duration of the intervention in their study [19].

Numerous systematic reviews have confirmed the effectiveness of myofascial release techniques in reducing pain, improving muscle function, enhancing performance, and promoting recovery [20,21,22]. However, two systematic reviews that analyzed the effects of myofascial release techniques on athletic performance were published a decade ago, and the results reported for ROM are still debated [23]. Potential methodological differences may limit the statistical power to evaluate the effects of myofascial release techniques on ROM performance, resulting in conflicting results in the existing literature. Inconsistent results may arise from methodological differences, which can restrict statistical power [24]. Therefore, a systematic review and meta-analysis approach may be preferred to ensure the generalizability of the findings and increase statistical power [25].

This systematic review and meta-analysis aimed to evaluate the effects of myofascial release techniques on ROM performance in athletes and to identify potential moderators that may regulate the effects of myofascial release techniques on ROM.

## 2. Materials and Methods

### 2.1. Registration of Systematic Review and Meta-Analysis Protocol

This systematic review and meta-analysis was conducted according to the *Cochrane Handbook for Systematic Reviews and Interventions* (version 6.3) guidelines [26] and was based on the Preferred Reporting Items for Systematic Reviews and Meta-Analysis (PRISMA) checklist statements [27]. The systematic review protocol was preregistered in PROSPERO (CRD42023429953), while the was protocol registered before data extraction started. All the study files are provided through the Open Science Framework (OSF) (https://osf.io/w48hs/, accessed on 15 April 2024).

### 2.2. Search Strategy and Selection of Studies

Two independent researchers performed the literature searches (A.B.A. and G.S.L.). In the case of discrepancies between the independent researchers, the discrepancies were resolved through discussions with a third researcher (A.P.). Electronic databases including the Cochrane Library, PubMed, Scopus, and Web of Science were searched to identify relevant articles published up to June 2023. Medical Subject Heading (MeSH) terms were examined to determine keywords, and the Word Frequency Analyzer Tool was used to suggest potentially relevant keywords (https://sr-accelerator.com/#/wordfreq, accessed on 7 July 2023). The following search strategy was adapted for each database: (Myofascial release [MeSH Terms] OR Myofascial Release Therapies [title] OR Therapy, Myofascial Release OR- Myofascial Release Treatments OR Treatment, Myofascial Release OR Myofascial Treatments OR Treatment, Myofascial) AND (range of motion[title] OR Joint ROM [title] OR flexibility [title]OR passive range of motion [title] OR active range of motion [title]) AND (athletes OR Professional Athletes [MeSH Terms] Or Elite Athletes [title] OR College Athletes [title/Abstract]). All the details about the coding strategy are provided in Appendix A. The eligibility of the studies that were exported from the relevant databases was evaluated using the Rayyan automation tool (https://www.rayyan.ai/, accessed on 7 July 2023). Two independent researchers (A.B.A. and G.S.I.) identified the studies that met our inclusion and exclusion criteria through peer-blinding using the Rayyan automation tool.

### 2.3. Eligibility Criteria

Participants, intervention, comparators, outcomes, and study design (PICOS) criteria were used to identify included and excluded studies. All the studies meeting the following criteria were included in this study: (i) healthy athletes who did not experience any injuries for at least three months, (ii) athletes who received myofascial release intervention, (iii) studies with active or passive control groups versus myofascial release intervention, (iv) ROM performance, (v) studies with a randomized controlled trial design, (vi) peer-reviewed articles, and (vii) studies that were published in English. The details of the PICOS criteria are provided in Table 1.

### 2.4. Data Extraction

The data extraction form was adapted from the electronic form in the Cochrane Handbook (https://dplp.cochrane.org/data-extraction-forms, accessed on 7 July 2022). The following data were extracted from the included studies: (i) study characteristics (authors, publication year, and study design), (ii) participant characteristics (age, gender, and type of athlete), (iii) type of intervention (intervention method used, duration of intervention, and details), (iv) measurement characteristics (ROM measurement method and joint type), (v) results (post-test mean and standard deviation of the experimental and control groups). The Tabula Data Extraction Tool was used for data extraction (https://tabula.technology/, accessed on 7 July 2023). If the authors reported the study results in the figures, a reliable software platform was preferred for the data extraction process (WebPlotDigitizer, version 4.5, https://automeris.io/WebPlotDigitizer/, accessed on 7 July 2023). The extracted data were reported using Microsoft Excel (Microsoft Corporation, Redmond, WA, USA). Data extraction was completed by three independent researchers (A.P., O.A., and H.S.U). The data extraction forms were cross-checked among the authors, and any differences were resolved by the fourth author (G.S.I.).

### 2.5. Assessment of the Methodological Quality of the Studies

Two independent researchers (O.A. and A.P.) evaluated the methodological quality of the included studies using the PEDro scale [28]. Two independent researchers scored the methodological quality of the included studies based on 11 quality criteria using the PEDro scale. The researchers checked whether the studies met each criterion in the PEDro scale. The methodological quality was interpreted according to the following reference values [29]: high quality (10–6 points), medium quality (4–5 points), and low quality (3–0 points). In case of a scoring discrepancy between the two independent researchers, a third researcher (G.S.I.) was involved in the methodological quality assessments. The third researcher resolved any disagreements in the evaluation.

### 2.6. Assessment of Evidence Quality

The studies’ certainty of evidence was assessed by two independent researchers (H.Ş.U. and A.B.A.) using the GRADE (Grading of Recommendations Assessment, Development, and Evaluation) scale. The meta-analysis results’ certainty of evidence was downgraded by one level for each of the following limitations: if the methodological quality assessment had a low level (*risk of bias*); if there was no significant difference in effect size results (*imprecision*); if the meta-analysis results indicated a high heterogeneity (*inconsistency*); if the ROM measurements were not performed using a valid and reliable method (*indirectness*); if publication bias was detected in the meta-analysis results and this bias could not be explained (*publication bias*) [30,31,32,33,34].

### 2.7. Statistical Analyses

The standardized mean difference was used to calculate the effect size (ES) and effect size variance between the myofascial release intervention and the active and passive control groups. The effect size of the studies was calculated using Cohen’s d effect size according to the following Equations (1) and (2) [35]:(1)$$Cohen\;d=\frac{M_{1}-M_{2}}{{SD}_{pooled}}$$
(2)$${SD}_{pooled}=\sqrt{\frac{\left(n_{1}-1\right)\times {SD}_{1}^{2}+\left(n_{2}-1\right)\times {SD}_{2}^{2}}{n_{1}+n_{2}-2}}$$

In Equations (1) and (2), M_1_ and M_2_ express the means of the groups, while SD_1_ and SD_2_ express the standard deviations of the groups. It also defines the sample size of the N1 and N2 groups and represents the pooled standard deviation of the SD_pooled_ groups. The effect size was interpreted according to the following reference values [29]: small effect (ES = 0.15), moderate effect (ES = 0.40), and large effect (ES = 0.75). A positive effect size indicated results favoring the myofascial release group versus the active or passive control groups. An a priori power analysis was performed to determine that the overall effect size analysis had sufficient statistical power. The power analysis was performed by applying the following criteria and establishing a two-sided hypothesis: (α = 0.05, β = 0.80, expected ES = 0.40, I^2^ = 75.00%, expected study size: 30, expected number of studies: 12). The results indicated that the myofascial release intervention should improve the athletes’ ROM performance with at least a moderate effect size (ES = 0.48).

On the other hand, meta-analytical packages assume that effect sizes are statistically independent [36]. However, most studies in the quantitative part of the synthesis (70%) reported two or more effect sizes to compare the effects of the myofascial release intervention on ROM performance. In this case, our study violated the rule of independence between effect sizes [37]. Therefore, the robust variance estimate (RVE) was used to calculate the overall effect size. Since the effect sizes were not completely correlated in a meta-analysis [38], the correlated and hierarchical effects model was used for the robust variance estimation method in the study. A RVE weighs effect sizes based on a predetermined within-study correlation and calculates the overall effect size by checking for the dependence of within-study effect sizes [39,40]. Researchers have also recommended RVE to control for small study effects [39]. The within-study correlation was set to ρ = 0.8 in the RVE. Sensitivity analyses were performed by changing the within-study correlation from ρ = 0.0 to ρ = 1.0. The inverse variance weight of each effect size was calculated according to Equation (3) [26]:(3)$$W_{ij}=\frac{1}{\kappa_{j}(\upsilon_{j}+\tau^{2})}$$

In Equation (3), $W_{ij}$ represents the inverse variance weight of the effect sizes. $\kappa_{j}$ is the number of effect sizes in each study used in the meta-analysis. $\upsilon_{j}$ denotes the mean of within-study sampling variances for $\kappa_{j}$ the effect sizes. τ^2^ explains the estimation of variance between the studies. The robust variance estimation was considered not to be reliable if the degrees of freedom (*df*) were four or less [36]. Possible heterogeneity was assessed using the τ^2^ and I^2^ index. The I^2^ value was interpreted according to the following reference values [41]: low heterogeneity (<25%), moderate heterogeneity (25–75%), and high heterogeneity (>75%).

The effects of outliers in the meta-analysis results were evaluated using Cook’s distance analysis. The overall effect size and publication bias analyses were repeated with and without outliers. A power-enhanced sunset funnel plot was used to control for publication bias. In addition, Egger’s regression test, Begg and Mazumdar’s test, and Rosenthal’s fail-safe N analysis were performed [42,43,44,45]. If the publication bias could not be explained, the analyses were repeated using the Trim and Fill method [46].

The following dependent variables in the study were considered as potential moderators and were coded categorically: (i) age, (ii) gender, (iii) duration of intervention, and (iv) type of intervention. The details of the coding categories for the subgroup analysis are presented in Appendix A. The subgroup analyses based on the RVE method were performed to identify potential moderators. The Hotelling Zhang test (HTZ) was performed to generate an F-value indicating whether different levels of moderators had varying effects on the ROM performance of the athletes. It was assumed that there was no statistically significant difference between the subgroups for the HTZ estimation. The cluster-robust (CR2) method was used for variance–covariance estimations.

All the statistical analyses of this study were completed using R version 4.1.0 (R Core Team). The effect size, effect size variance, and publication bias analyses were calculated using the {metafor} package. Power analysis was performed using the {metapoweR} package. Outliers were determined using the {dmetar} package. The power-enhanced sunset funnel plot was created using the {metavis} package. The HTZ analysis was performed using the {clubsandwic} packages. Robust variance estimations and subgroup analyses were performed using the {robumeta} package. RVE codes were adapted through previous studies [36,47]. The statistical significance level was set at *p* ≤ 0.05 in all the analyses.

## 3. Results

### 3.1. Study Selection

The literature search identified 411 studies from four electronic databases. After removing the duplicates, the titles and abstracts of 340 studies were evaluated. Out of the 340 studies, 44 were screened in full-text. Ten studies were considered that met the PICOS criteria for eligibility, and these studies were included in the systematic review and meta-analysis. The details of the literature search and study selection are provided in Figure 1.

### 3.2. Characteristics of the Included Studies

The systematic review and meta-analysis included ten studies. The pooled effect size of the ten included studies was 282. The study sample sizes ranged from 24 to 67 athletes [48,49]. Two of the studies included both genders [50,51]. Seven studies included exclusively male athletes, while one focused on female athletes [49]. The average age of the athletes ranged from 11.87 to 29.3 years [51,52]. The youngest athlete was 10 [52], while the oldest was 48 [51]. The participants were competitive swimming (n = 30), soccer (n = 123), rugby (n = 20), handball (30), and track and field athletes (n = 79). The ROM was measured using an inclinometer [49,50,53,54], a goniometer [52,55,56], the Biodex Multi-Joint System 4 Pro [48], and a stand-and-reach test device [51]. Five of the studies followed the acute period of the intervention session [49,51,53,54], while the other five followed the long-term period [48,52,55,56]. The duration of a single myofascial intervention session ranged from 1 min and 20 s [56] to as long as 40 min [52]. The characteristics of the interventions consisted of instrument-assisted manual therapy (IAMT) [49], foam rolling [50,51,56], neurodynamic sliding techniques [57], fascial manipulation [55], dry needling [53], and dry needling combined with a water pressure massage [54]. The study characteristics are detailed in Table 2.

### 3.3. Evaluation of Methodological Quality

In the methodological quality assessment, the studies were scored, ranging from 6 to 8. The pooled PEDro quality score for all the studies was calculated to be 7.1. This score showed that the studies had a high quality in terms of methodology, and no studies were excluded due to methodological quality. The ten studies included in this study received one point for each of the following criteria: (i) randomized allocation, (ii) between-group comparisons for at least one key outcome, and (iii) point and variability measures for at least one key outcome. The details of the methodological quality assessment are presented in Appendix A.

### 3.4. The Results of the Overall Effect Size

The overall effect size analysis was performed using 78 effect sizes from the 10 studies. The myofascial release group comprised 983 participants, while the active or passive control groups comprised 974 participants. The meta-analysis results indicated that the myofascial release intervention has a moderate effect on ROM performance in athletes compared to the active or passive control groups (ES = 0.53, 95% CI = 0.18 to 0.89, *p* = 0.01). Since the *df* was greater than four, the results of the meta-analysis based on the dependent effect size were accepted as reliable (*df* = 8.96). Moreover, the sensitivity analysis showed that the assumed within-study correlation did not affect the results of the overall effect size analysis (Appendix A). Due to the high heterogeneity in the meta-analysis results (I^2^ = 77.60%, τ^2^ = 0.59), we performed an outlier analysis and moderator analysis to identify potential sources of heterogeneity. The details of the forest plot for the overall effect size are presented in Figure 2.

### 3.5. The Results of the Moderator Analyses

#### 3.5.1. Age

Age had a moderating effect on myofascial release on the ROM performance of athletes (*F* = 1.62, *df* = 2, *p* = 0.01). A subgroup analysis revealed that myofascial release intervention at ≤18 years of age produced a significantly greater effect on the ROM performance of athletes compared to the >18 age group (ES = 1.10, 95% CI = 0.47 to 1.72, *df* = 1.00, *p* = 0.02). However, it was understood that the results might not be reliable due to the df value being lower than the threshold value.

#### 3.5.2. Gender

Gender had a moderating effect on the ROM performance of the athletes (*F* = 12, *df* = 3.28, *p* = 0.03). A subgroup analysis showed that the myofascial release intervention in male athletes produced a significantly greater effect on the ROM performance compared to the female and mixed groups (ES = 0.77, 95% CI = 0.41 to 1.12, *df* = 1.00, *p* = 0.01). However, the gender moderator was not considered reliable because the *df* value was lower than the threshold value.

#### 3.5.3. Intervention Duration

The intervention duration revealed a moderating effect on the ROM performance of the athletes (*F* = 9.80, *df* = 4.98, *p* = 0.01). Long-term myofascial release intervention improved the ROM performance of the athletes to a greater extent compared to short-term applications (ES = 0.71, 95% CI = 0.28 to 1.14, *df* = 3.99, *p* = 0.01). Since the *df* value was higher than the threshold value, the moderator effect of the intervention time was considered reliable.

#### 3.5.4. Joint Type

The joint type displayed a moderating effect on the ROM performance of the athletes (*F* = 1.16, *df* = 3.22, *p* = 0.01). The myofascial release intervention had a significantly greater effect on cervical joint ROM compared to the other joints (ES = 1.04, 95% CI = 1.05 to 1.05, *df* = 1.00, *p* = 0.01). However, the joint type moderator was not considered reliable because the *df* value was lower than the threshold value.

#### 3.5.5. Intervention Type

The intervention type showed a moderator effect on the ROM performance of the athletes (*F* = 4.63, *df* = 14.00, *p* = 0.01). The other intervention methods improved the athletes’ ROM more than the instrumental and myofascial release interventions (ES = 1.05, 95% CI = 1.05 to 1.05, *df* = 1.00, *p* = 0.01). The intervention type produced reliable results regarding the moderator effect. However, it was not considered reliable for subgroup analysis because the *df* value was lower than the threshold value. The details of the subgroup and moderator analyses are presented in Table 3 and Figure 3.

### 3.6. The Results of Publication Bias Analyses

The power-enhanced sunset funnel plot showed that the studies had a low statistical power (β = ≤55%), and there may have been publication bias in the overall effect size results (Figure 4). In addition, the three bias analyses indicated publication bias regarding the overall effect size. Therefore, the overall effect size was recalculated using the trim-and-fill method. The trim-and-fill analysis results showed that publication bias did not affect the overall effect size level (ES = 0.34, 95% CI = 0.20 to 0.48, *p* = 0.01). The detailed results of publication bias are presented in Table 4 and Figure 4.

### 3.7. The Effects of Outliers on the Analyses

A Cook’s distance analysis determined that there were five outliers in the overall effect size results (Appendix A). After removing the outliers, the overall effect size was recalculated using 73 effect sizes from 9 studies. The results revealed that the outliers did not have an impact on the overall effect size (ES = 0.44, 95% CI = 0.12 to 0.77, *df* = 7.81, *p* = 0.01). Similarly, the publication bias analyses were evaluated without outliers, and the presence of outliers did not impact the assessment of publication bias. On the other hand, the outliers were a potential source of heterogeneity. After removing the outliers, we found moderate heterogeneity in the overall effect size analysis (I^2^ = 49.00%).

### 3.8. Certainty Assessment and Power Analysis

A high heterogeneity was observed in the overall effect size and moderator analyses (I^2^ > 75%). Therefore, the certainty of the evidence was downgraded by one level, and the results of the meta-analysis were reported as being of moderate quality. Thus, it was assumed that the overall effect size and moderator analysis results were close to the actual effect. The details of the certainty of the evidence are presented in Appendix A.

On the other hand, it was determined that the overall effect size was similar to the power analysis results (estimated ES = 0.40; actual ES = 0.53). Thus, it was revealed that the meta-analysis results had at least β = 80% statistical power.

### 3.9. The Adverse Effects Reported by Myofascial Release Interventions

The included studies did not report adverse effects or injury for the myofascial release methods. According to these results, it can be said that the myofascial release method is a reliable method for athletes.

## 4. Discussion

This systematic review and meta-analysis aimed to evaluate the effects of myofascial release techniques on the ROM of athletes and identify potential moderators. Previous studies have evaluated the effects of MRT on body pain. Additionally, although the impact of MRT on ROM was assessed, the study population consisted of sedentary individuals. Therefore, the current study aimed to present a unique approach to the literature by evaluating the effects of myofascial release techniques on athletes’ ROM performance. The results showed that the myofascial release techniques had a moderate effect size on the ROM of the athletes compared with the active or passive control groups (ES = 0.53, 95% CI = 0.18 to 0.89, *p* = 0.01).

After analyzing other systematic reviews, it was confirmed that a single, manually applied myofascial technique can improve ROM without negatively impacting muscle performance in the short term [58,59]. Our study found that the effects of myofascial techniques on ROM can be maintained over time and through multiple sessions. This contrasts with the findings of previous studies [14,59], which did not report a sustained increase in ROM over time.

According to the literature, the improvement in the ROM of athletes can be attributed to various factors, including neurological, mechanical, physiological, or functional reasons. The effects of myofascial release techniques on ROM can be explained by the changes they induce in the arterial, venous, and lymphatic circulation systems [60,61]. Myofascial release can improve blood circulation, bringing oxygen and nutrients to the tissues and facilitating metabolic changes between the fascia and the extracellular matrix [60,61]. Tissue alkalinization could reduce pain and improve ROM by releasing muscle tension and soreness [62]. Another critical factor that may trigger the development of ROM is tissue hydration [63]. Since most of the ground substance is comprised of water [64], which plays a crucial role in determining the stiffness of the myofascial system [65], the water content within the fascia can have an impact on the ROM level. In addition to the already-mentioned physiological mechanisms, the mechanical pressure applied during the myofascial release technique can have a positive impact on collagen cross-link adhesions and fibroblast cell activity [66,67]. Fascial adhesions broken down during myofascial release work are one of the main short-term physiological effects of myofascial release techniques. In the long term, laboratory experiments demonstrate that fibroblasts, the primary cells of fascial tissue, adapt specifically to mechanical loading. This adaptation depends upon the stimulus’s strain, duration, and frequency [68]. Extracellular matrix reorganization, produced by fibroblast cells, can be the primary mechanism for maintaining gains in joint ROM. This mechanism can be explained by a property called plasticity [20]. The effects of myofascial release intervention on ROM may also be based on neurological mechanisms. Langevin proposed that fascia functions as a body-wide signaling network [69]. Since the fascia can “feel” due to the presence of mechanoreceptors, tissue manipulation leads to inhibition of the sympathetic nervous system and muscle tonus [64,70]. These two neurological mechanisms can explain the effects of myofascial release interventions, which do not have a strong mechanical impact, such as neurodynamic sliding techniques or dry needling [71].

This systematic review and meta-analysis included interventions such as foam rolling, IAMT, neurodynamic shift techniques, facial manipulation, and dry needling. Our results showed that the interventions based on myofascial trigger point therapy and PNF contraction–relaxation techniques had larger effect sizes (ES = 1.05, 95% CI = 1.05 to 1.05, *p* = 0.01) compared to myofascial release methods (ES = 0.41, 95% CI = −0.18 to 1.01, *p* = 0.13) and instrumental methods (ES = 0.61, 95% CI = −3.43 to 4.65, *p* = 0.30). The nervous system hypothesis can support this result because these two types of myofascial release can effectively stimulate Golgi tendon organs and muscle spindles, thereby inhibiting muscle tone and further increasing joint ROM [72,73]. It may be possible to improve the ROM by manipulating specific muscle groups within a myofascial chain, working on interconnected muscle chains, and considering the potential tension transfer between them [74]. The evidence for myofascial chains is limited based on the anatomical connections between muscles, bones, and ligaments [20]. Therefore, studies are needed to determine the mechanical significance of myofascial chains [75]. The myofascial release intervention performed on athletes ages 18 and under resulted in greater development of their ROM (ES = 1.10, 95% CI = 0.47 to 1.72, *p* = 0.02) compared to those above the age of 18 (ES = 0.40, 95% CI = 0.03 to 0.77, *p* = 0.02). These findings can be attributed to various factors, including the elastic properties of soft tissue, the turnover and quantity of collagen cells, fascial adhesions, and the plasticity of the nervous system [76,77,78,79]. These factors are generally more favorable in younger individuals. We also found that longer interventions (ES = 0.71, 95% CI = 0.28 to 1.14, *p* = 0.01) had a better outcome in terms of the ROM level compared to shorter interventions (ES = 0.37, 95% CI = −0.34 to 1.10, *p* = 0.22). The properties of fascial tissue, such as viscoelasticity, plasticity, and thixotropy, can explain these results [80,81]. Also, other factors such as tissue hydration, ECM reorganization, and nervous system plasticity can contribute to long-term improvements in the joint ROM of athletes. Regarding joint localization, a statistically significant improvement was found in the ROM at the cervical spine level (ES = 1.04, 95% CI = 1.05 to 1.05, *p* = 0.01). The myofascial release intervention had a positive impact on other joints (shoulder, knee, ankle), but it did not reach statistical significance.

The meta-analysis results of this study can provide field experts and researchers with a perspective on the subject. However, only the type of intervention and the corresponding outcomes are highly reliable among the meta-analysis results. Researchers have reported that a minimum of four degrees of freedom is necessary for obtaining reliable results in studies based on a robust analysis of variance [36]. On the other hand, it was observed that the current findings exhibited a high heterogeneity. Therefore, the certainty of the evidence was reduced by one level. The statistical cause of heterogeneity may be outliers. Researchers have stated that outliers may cause heterogeneity [82]. In our study, after removing outliers from the dataset, the level of heterogeneity in the results decreased. This situation may support the stated hypothesis. In addition, another source of heterogeneity may be related to the study methodology. Researchers have reported that methodological differences in studies may also cause heterogeneity [83]. Since many methods were used, such as MRT, in this study, methodological differences may have contributed to the heterogeneity of the results.

### Limitations

The dependent effect sizes were controlled in this study, and the results were reported with a statistical power of at least 80%. The outliers did not affect the overall effect size and publication bias analyses. However, it has been determined that outliers can contribute to heterogeneity. It was understood that controlling for publication bias would yield similar results to the overall effect size results. The included studies were of good methodological quality, while the certainty of evidence from the meta-analysis results was reported as moderate level. There were some limitations in the systematic review and meta-analysis. Firstly, the variety of intervention methods may have contributed to the moderate heterogeneity. Secondly, the athletes’ ROM performance was measured at different joints. This is a limitation in interpreting the effects of myofascial release on a single ROM. Based on eight effect sizes extracted from one study, it was found that the cervical joint showed a more significant improvement in the ROM compared to the other joints. However, outliers may impact effect sizes, since these results were obtained from a single study. This case may have influenced the results. In addition to these limitations, the studies included athletes from various sports. Therefore, it is unclear which sports benefit the most from myofascial release techniques. Gender factors may also limit the effectiveness of myofascial release techniques because a single study assessed the effects of myofascial release on the ROM of female athletes. Although the effects of myofascial release techniques compared to active or passive control groups were often investigated in the present literature, there were few studies on the impacts of different intervention methods. Therefore, the myofascial release techniques were combined to determine the best effects/improvements in the ROM. The diversity of MRI techniques and comparison groups limits the definitive interpretation of the effects of this technique on ROM. Therefore, there is a need to re-investigate the effects of MRT on ROM using similar research protocols. Thus, the effects of MRT on ROM can be determined more clearly by comparing the control and intervention groups.

## 5. Conclusions

Myofascial release techniques may further improve athletes’ range of motion compared to active or passive control groups. The ROM level of younger athletes may be enhanced with MRTs, and both genders may derive similar benefits from the effect of MRTs on ROM. With MRTs, the ROM level of the cervical region can be improved versus other joint types. Regarding MRTs, techniques such as dry needling may improve ROM. Although the study results confirm the effectiveness of MRTs on the ROM of athletes, the current evidence does not have a high certainty. Therefore, future studies with similar protocols are needed. Thus, a higher certainty of evidence can be obtained for the effects of MRTs on the ROM of athletes.

## Figures and Tables

**Figure 1 sports-12-00132-f001:**
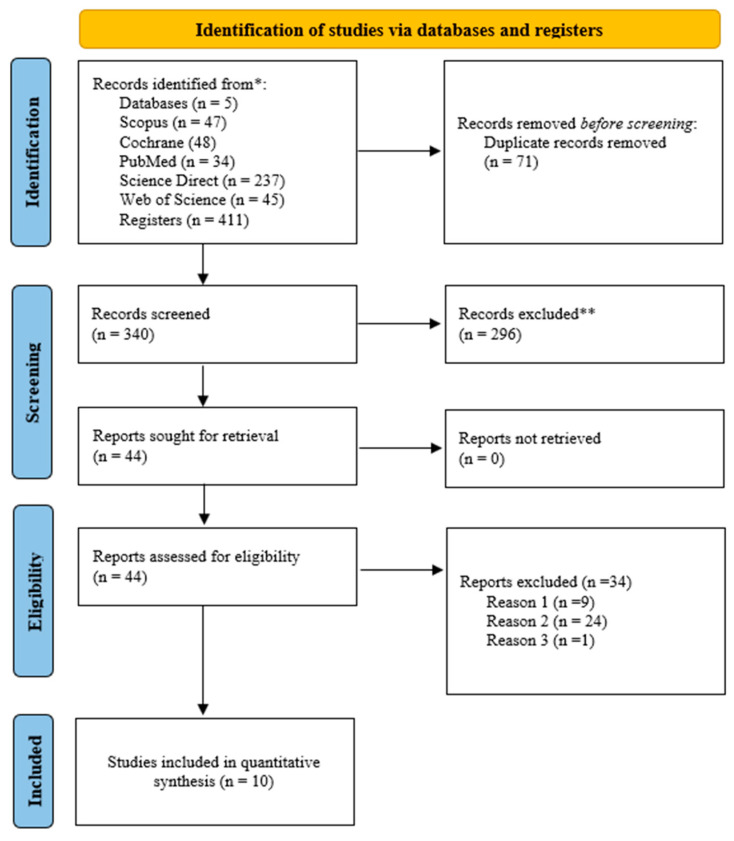
The literature search and study selection based on the PRISMA for systematic review and meta-analysis. (Reason 1) Study subject does not meet PICOS criteria; (Reason 2) study design not appropriate; (Reason 3) athletes not included in the study. *: Electronic databases; **: Studies excluded by title and abstract review.

**Figure 2 sports-12-00132-f002:**
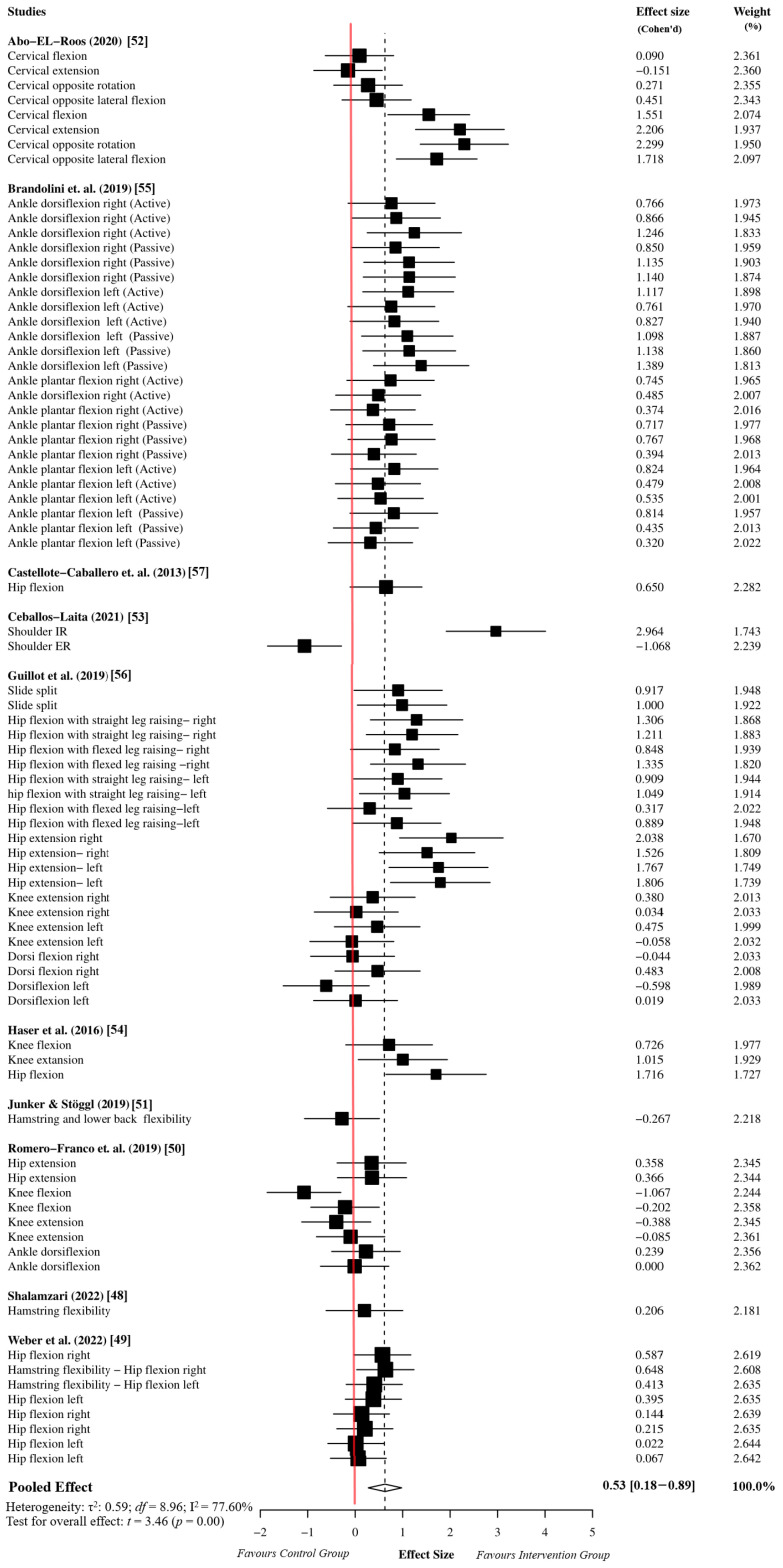
Overall effect size analysis results based on robust variance estimation [48,49,50,51,52,53,54,55,56,57].

**Figure 3 sports-12-00132-f003:**
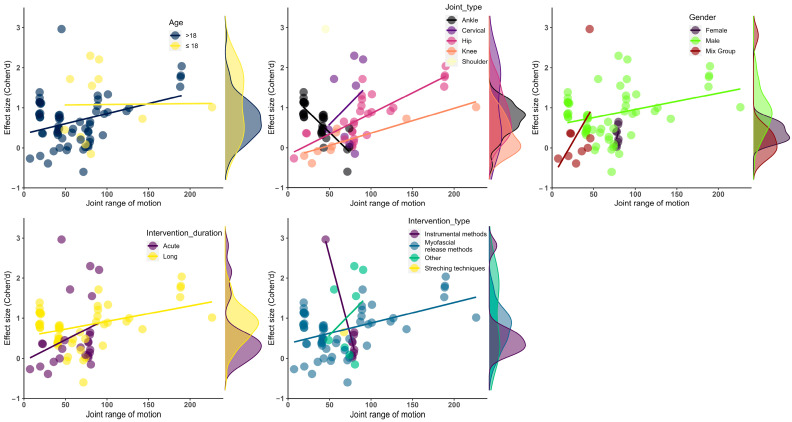
Visualization of effect size results based on mean ROM. (**up-left**) Age moderator; (**up-middle**) joint type moderator; (**up-right**) gender moderator; (**bottom-left**) intervention duration moderator, (**bottom-right**) intervention type moderator.

**Figure 4 sports-12-00132-f004:**
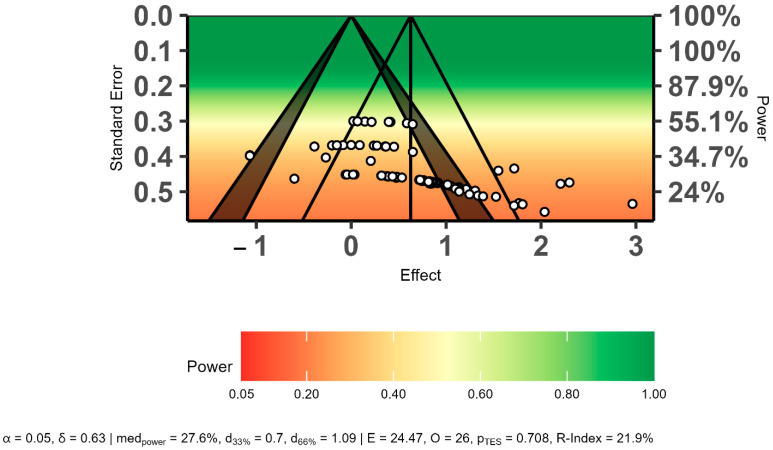
Evaluation of publication bias using a power-enhanced sunset funnel plot.

**Table 1 sports-12-00132-t001:** Study inclusion and exclusion criteria based on PICOS.

	Inclusion Criteria	Exclusion Criteria
**Population**	Male and female athletes who are healthy, actively participate in training and competitions, and have not experienced any injury in the last three months	Male and female athletes with health problems, active non-athletes, or injuries
**Intervention**	Studies using any type of myofascial release interventions, either acute or long-term	Studies combining different techniques with myofascial release intervention. Myofascial release interventions applied biweekly or intermittently
**Comparator**	Myofascial release intervention group versus a control group (active or passive)	Studies without an adequate comparator group (e.g., single-group study designs; cross-over sectional studies)
**Outcome**	Studies evaluating the effects of myofascial release intervention on ROM	Studies evaluating the effects of myofascial release interventions apart from ROM
**Study design**	Randomized controlled trials	Single-arm study designs or non-randomized controlled trials
**Additional criteria**	Full-text original peer-reviewed articles scoring four or higher on the PEDro scale	Articles scoring three or lower on the PEDro scale, theses, unpublished articles, reviews

**Table 2 sports-12-00132-t002:** The characteristics of the included studies.

Authors/Year	Age (Mean ± SD)	Gender	Type of Athletes	The Characteristics of Intervention Time	Intervention Type	The Details of Intervention	ROM Measurement Tools
Weber et al. (2022) [49]	20.9 ± 3.9	Female	Soccer players	8 min	IAMT	Metabolization (shirt frictions) and rehydration (slowly moving through the tissue, pushing a shifting skin fold) were performed with the Faser 2 tool.	Inclinometer (AcuAngle, Baseline, Elmsford, New York, NY, USA)
Junker & Siegel (2019) [51]	29.3 ± 8.5	Male + Female	Recreational active athletes	8 weeks/16 session	Foam rolling	The foam rolling was positioned on calf muscles, quadriceps femoris, hamstrings, iliotibial band, and gluteal muscles.	Stand and reach test vehicle
Guillot et al. (2019) [56]	20.6 ± 0.08	Male	Rugby players	6 weeks/15 sessions	Foam rolling	SMR 20 and 40 s (hip extensors, hip adductors, knee extensors, knee and plantar flexors).	Electronic goniometer (MLTS700, Australia)
Castello-Caballero et al. (2013) [57]	20.7 ± 1.0	Male	Soccer players	1 week/3 session	Neurodynamic sliding techniques	Seated straight leg sliders	Passive SLR test- plastic goniometer
Brandolini et al. (2019) [55]	29.00 ± 8.58	Male	Soccer players	3 weeks/3 session	Fascial manipulation	Deep friction over specific points: Centre of Coordination and Centre of Fusion	Universal goniometer with two arms
Romero-Franco et al. (2019) [50]	24.55 ± 4.45	Male + Female	Athletes	6 min	Jogging + Foam rolling	8 min of jogging + 4.5 Gs foam rolling (posterior thigh, from the popliteal fossa to the ischial tuberosity; anterior thigh, from the anterior-superior iliac spine to the quadriceps tendon; and calf, from the popliteal fossa to the Achilles tendon).	Modified Thomas test- mini digital inclinometer (GO 90532, Sweden)
Abo-El-Roos (2020) [52]	11.87 ± 1.36	Male	Young Swimmers	3 months/36 session	Other methods	Physical therapy program + lidocaine hydrochloride gel + phonophoresis	Goniometer
Ceballos-Laita (2021) [53]	22.39 ± 3.73	Male + Female	Handball players	N/A	Dry needling	A single session of DN guided by ultrasound into active MTrPs in the rectus femoris muscle was placed in a supine position.	Digital inclinometer
Haser et al. (2016) [54]	18.4 ± 2.5	Male	Soccer players	4 weeks/20 sessions	DN with water pressure massage placebo laser with water	Acupuncture needles were inserted into TP of the front and back of the athletes’ thighs. When the needle elicited a local twitch response, it was removed.	Plurimeter (Baseline, Bubble inclinometer)
Shalamzari (2022) [48]	24.91 ± 1.98	Male	Active athletes	3 times per week/ 8-week	Self-myofascial release (foam rolling)	The subject was asked to use a foam roller that consisted of a 6-in diameter × 15-in length foam roller for the hamstring muscles. Then, the subject was instructed to support the body weight and roll up and down for 2 min.	Biodex MultiJoint System 4 Pro dynamometer

Note: SMR: Self myofascial release; IAMT: Institute of advenced muscoskeletal treatments; DN: Dry needling.

**Table 3 sports-12-00132-t003:** The results of moderator and subgroup analyses.

The Effect Size of the Subgroups								Moderator Test			
Variables	N_study number_	N_ES number_	ES	95% CI	*t*	*df*	*p*-Value	*HTZ* Statistic *(F)*	*df*	*p*-Value	I^2^
**Age**								*F* = 1.62	2	0.01 *	77.00%
≤18 years of age	2	11	1.10	0.47 to 1.72	22.52	1.00	0.02 *				
>18 years of age	8	67	0.40	0.03 to 0.77	2.57	6.97	0.03 *				
**Gender**								*F* = 12	3.28	0.03 *	79.00%
Male	6	59	0.77	0.41 to 1.12	5.57	1.00	0.01 *				
Female	1	8	0.31	0.31 to 0.31	70.00	4.99	0.01 *				
Mix group (male + female)	3	11	0.17	−1.40 to 1.76	0.48	2.00	0.67				
**Intervention duration**								*F* = 9.80	4.98	0.01 *	78.00%
Acute	5	27	0.37	−0.34 to 1.10	1.44	3.99	0.22				
Long	5	51	0.71	0.28 to 1.14	4.63	3.99	0.01 *				
**Joint type**								*F* = 1.36	3.22	0.01 *	82.00%
Ankle	3	30	0.59	−1.48 to 2.62	1.83	1.41	0.25				
Knee	3	10	0.26	−2.79 to 3.32	0.50	1.55	0.67				
Hip	7	28	0.43	−0.09 to 0.96	2.11	4.87	0.08				
Cervical	1	8	1.04	1.05 to 1.05	30.00	1.00	0.01 *				
Shoulder	1	2	N/A								
**Intervention type**								*F* = 4.63	14.00	0.01 *	83.00%
Instrumental methods	2	10	0.61	−3.43 to 4.65	1.92	1.00	0.30				
Myofascial release methods	5	59	0.41	−0.18 to 1.01	1.79	4.99	0.13				
Other methods	1	8	1.05	1.05 to 1.05	65.00	1.00	0.01 *				
Stretching methods	1	1	N/A								

Note 1: ES: Effect size; HTZ: Hotelling Zang test; N/A: not available; 95% CI: The upper and lower limits of the confidence interval; *: *p* < 0.05. Note 2: The results were considered unreliable if the degrees of freedom (*df*) were four or less.

**Table 4 sports-12-00132-t004:** The results of publication bias analyses in the study.

Publication Bias Analysis	ROM Performance
Rosenthal Fail-Safe N (*p*-value)	4496 (0.0001) *
Rosenthal Fail-Safe N without outliers (*p*-value)	3742 (0.0001) *
Egger’s regression test (t-value, *p*-value)	6.02 (0.01) *
Egger’s regression test without outliers (t-value, *p*-value)	6.49 (0.01) *
Begg and Mazumdar Test (z-value, *p*-value)	0.69 (0.01) *
Begg and Mazumdar Test without outliers (z-value, *p*-value)	0.71 (0.01) *
Trim-and-fill method (ES [95%CI], *p*-value)	0.34 (0.20–0.48) *p* = 0.01

Note: ES: Effect size; ROM: joint range of motion; 95%CI: The lower and upper limit of a 95% confidence interval; * *p* < 0.05.

## Data Availability

The datasets generated and/or analyzed during the current review are available in tables, Appendix A, and at the Open Science Framework (https://osf.io/w48hs/).

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
