# Peer review of "Effects of Myofascial Release Techniques on Joint Range of Motion of Athletes: A Systematic Review and Meta-Analysis of Randomized Controlled Trials"

_sports, 2024, doi:10.3390/sports12050132_

Round 1
Reviewer 1 Report
Comments and Suggestions for Authors
This is an interesting article. Congratulations to the authors, I think it is a meta-analysis that addresses an important topic, but it has some flaws in the methodology compared to what we want to conclude.
In the introduction it is not possible to use the abbreviation ES without first indicating its meaning; Effect size.
On line 42, once you use the abbreviation ROM on the first line, you will no longer have to enter range of motion.
It is necessary to revise the introduction, without using numerical characters, emphasizing the need for this revision.
The discussion and conclusion has evidence that is not precise, such as coming to the conclusion that it may be good for myofascial trigger points, but the goal of this study is to see the effect of myofascial therapy, without trigger point.
I don't understand the point of including studies with foam rolling, dry needling, and other techniques, saying that the results are compared to the control group, when not all groups use controls. Including all these types of study, and analyzing these variables per study does not make sense, it would have to be divided by zones or by techniques for its analysis to make sense.
The meta-analysis only addresses per study, without comparing post-treatment results with other therapies or with other studies, this makes it a less solid meta-analysis.
Author Response
Dear reviewer, thank you for spending your time to review our work. The changes made in the manuscript are presented in the word document that we uploaded.

Reviewer 2 Report
Comments and Suggestions for Authors
The aim of this interesting systematic review and meta-analysis is to evaluate the effect of myofascial release techniques on the ROM performance of athletes.
Abstract: Please include the databases used in the study.
Introduction: After the sentence "ROM can impact various fitness components that require strength, power, sprint speed, and vertical jump [2, 3]," add that clinical evaluation of musculoskeletal system, and sometimes preventive assessment (clinical or with non-invasive methods, such as US), is essential in athletes to maintain their performance and add references (Musculoskeletal ultrasound is an increasingly effective and repeatable diagnostic tool when performed using precise landmarks (e.g., pennation angle), and in athletes, it can also be used as screening to prevent subsequent injuries, as in overhead athletes (e.g., volleyball players).).
Results: Add a table summarizing the included articles (authors, participants, cause of treatment, characteristics of treatment, results...).
Discussion: At the end of "This systematic review and meta-analysis aimed to evaluate the effects of myofascial release techniques on the ROM of athletes and identify potential moderators. The results showed that myofascial release techniques had a moderate effect size on the ROM of the athletes compared with the control group (ES = 0.53, 95% CI = 0.18 to 0.89, p = 0.00)," please provide more precise p-value details.
These proposed concepts are presented as suggestions to enhance the readability and content of the text. Their inclusion is recommended for a more comprehensive and well-supported discussion. The references are suggested based on the reviewer's knowledge of the content relevant to this study. However, the authors are encouraged to explore additional references that may be also pertinent to their work.
I believe that addressing these points will significantly enhance the clarity and completeness of your studies. We look forward to seeing the revised manuscript.
Comments on the Quality of English LanguageThe text is enough readable and fluid.
Author Response

(The authors gave the same response as above.)

Reviewer 3 Report
Comments and Suggestions for Authors
The present study aims to evaluate the effect of myofascial release techniques on athletes' range of motion performance. The results highlighted that the myofascial release intervention had a moderate effect on range of motion performance in athletes when compared to the control group.
I have several suggestions.
1. Line 46-47: "Optimal ROM can enhance the performance of athletes. However, the ROM of an athlete may be restricted by a specific activity [9], biological factors [10], or anthropometric characteristics [11]". The introduction section lacks a detailed description of the factors that may influence the limitation of ROM in athletes. It would be worth pointing them out so they can be referred to later in the paper.
2. Please improve the writing equations in the text (line 153) according to the paper's journal requirements.
3. It would be worth also describing the methods of self-myofascial therapy from the works included in the review regarding effectiveness in the discussion section.
In general, the work is interesting and can contribute to the literature. I hope my suggestions will help improve this work.
Author Response

(The authors gave the same response as above.)

Reviewer 4 Report
Comments and Suggestions for Authors
INTRODUCTION
There has been a number of systematic reviews on the subject, some of them including a meta-analysis. The authors did mention them, but the fail to justify their study. There is a number of reasons for performing a new systematic review and meta-analysis: Methodological flaws detected/ need for an update since the search was performed years ago/need for an update since new studies have been published since the las review published/need for study a specific population.
The authors state none of these reasons. It can be guess that they justify their study on the bases of contradictory findings but this is not clear and not enough.
Justification for performing this systematic review should be stronger
METHODS
This section is well described and accurate. My main concern is why the authors used PEDro (which is known to have several weaknesses for assessing interventions in which participants and also evaluators are difficult to be blinded) instead of Cochrane Risk of Bias criteria.
RESULTS
Figure 1 does not show the latest version of the PRISMA flow chart, which provides information on references located through citation search and reference search.
Also, I am afraid the item “report for included studies” is not accurate. It can not be possible that all studies shared the same sample characteristics.
In addition, it should be stated how many studies were included in the meta-analysis.
3.2. How many studies included: professional athletes, college athletes and elite athletes?
Which were the main activities performed by the comparison groups?
Ten studies were found. A table providing sample characteristics (including number of participants in each study), main intervention and comparison intervention, main outcomes, and results (inter and intragroup) should be shown in the text, instead of Appendix.
Methodological Quality: How many studies were directly retrieved from PEDRO?
Are the authors confident on perform a moderator analysis on several variables (age, duration intervention….) with only ten studies???
DISCUSSION
What are the differences between this present review and other reviews on the topic?
In which way this review updates or improves the scientific evidence reported in previous reviews?
CONCLUSION
Too long and almost redundant in places. How strong is the scientific evidence on the effects of myofascial release? Is there a need for future studies? Which methodological flaws should overcome such studies? Which myofascial release techniques seems to be better than others?
Comments on the Quality of English LanguageEnglish is fine
Author Response

(The authors gave the same response as above.)

Reviewer 5 Report
Comments and Suggestions for Authors
This is an impressive study, but it has some serious drawbacks.
line 5: validate George Sebastian lacob
Abstract. Correct the abstract following the comments below.
Intorudction. The topic is well justified.
Methods. The selection of material and statistical methods are sufficiently well described.
Results. Consider whether it is worth splitting the analysis into a section regarding the ranges of motion for the spine joints and limb joints. It is worth noting that the training, loads and muscle size are different and the goal resulting from increasing the range of motion is different. In such a case, it is worth dividing Figure 2 into two separate ones. Also, myofascial release flow analysis should be calculated separately.
In the results, you provide the probability level p = 0.00. Such a result would indicate no probability - use p < 0.001.
Table 2 and selection of moderators. Justify the choice of groups <25 and >=25. Such a division has no justification in sports. The only significant myofascial release effect is for cervical. Be critical of this result. You also assessed the impact of intervention type. Consider adding a chart specifying the effect test values for the impact of the intervention type, such as in Figure 2.
Discussion. In the discussion, you refer only to the results of moderator and subgroup analyzes and only in the scope of the moderator test. You do not address the effect size of subgroups. The lack of this reference is a mistake because the effect size of subgroups suggests moderator effects resulting from, for example, the significance of only one variable. Critically analyze the results of the statistical analysis.
Conclusions. It is impossible to agree with the conclusions. Why do you say that younger athletes can achieve greater ROM development with myofascial release techniques. Gender, intervention duration, and joint type may also have a moderating effect on the effectiveness of a fascial release intervention. As for gender, the effects were the same for both genders, and age showed effects only in the younger group under 25 years of age. But why was such a division into age groups made? The results regarding the effect on range of motion were significant only for cervical. How do you justify the conclusion that the effects on the other joints were not significant?
Author Response

(The authors gave the same response as above.)

Round 2
Reviewer 2 Report
Comments and Suggestions for Authors
I appreciated the efforts of the authors to improve the article according to my suggestion.
Comments on the Quality of English Languageminor revision
Author Response
Thank you!

Reviewer 4 Report
Comments and Suggestions for Authors
I have read all the answers provided by the authors. I only have one request.
The authors should state how many RCTs were directly retrieved from PEDro. It is not clear whether they assessed the methodological quality of the manuscript or the score was directly extracted from PEDro.
Comments on the Quality of English LanguageNo comments.
Author Response
Thank you!

Reviewer 5 Report
Comments and Suggestions for Authors
All comments in the text have been taken into account. This version appears to be suitable for publication.
Author Response
Thank you!
